# VIZAGENTBENCH:
# BENCHMARKING MULTIMODAL AGENT REASONING ON MULTI-VIEW VISUAL ANALYTICS

## ABSTRACT

Multimodal Large Language Models (MLLMs) can now act as full-fledged desktop agents, yet their visual reasoning skills remain largely evaluated on single, static charts. Real decision making, however, happens in dashboards that combine multiple coordinated views (MCVs) and rely on rich interactions such as brushing, filtering, and drilling down. We introduce **VizAgentBench**, the first benchmark that challenges agents to perceive screenshots of a live MCV dashboard, issue declarative interaction commands, and answer analytical questions whose solutions may be hidden behind dynamic tooltips or axis changes. VizAgentBench is constructed by (1) surveying 14 visualization research papers to derive a design space of chart-and-interaction templates; (2) mining 10 public Kaggle datasets across finance, healthcare, sports, and socio-economics; and (3) generating 192 dashboards paired with same number of question–answer tasks using a large language model plus manual validation by a group of graduate students in data science. On our benchmark, state-of-the-art LLM agents achieve only ∼40% accuracy, revealing substantial headroom. We release the dashboards, data, and an open-source API that separates perception from action, lowering the barrier to agent research on interactive visualization.[1]

## 1 INTRODUCTION

Large multimodal language models (MLLMs) are rapidly evolving from passive image-captioning systems into *interactive agents* that can perceive an entire desktop, ground their reasoning in visual context, and execute mouse-and-keyboard actions. OpenAI's Computer-Using Agent (CUA) and Anthropic's "computer-use" mode already automate multi-step office workflows such as spreadsheet reconciliation, slide editing, and website navigation (OpenAI, 2025a; Anthropic, 2024). A critical—but so far under-examined—capability for these GUI agents is the ability to read and *interact with data visualizations*. Business-intelligence analysts, journalists, and scientists routinely pivot, filter, and link multiple coordinated views (MCVs) to discover patterns and drive decisions (Keim, 2002). Agents that cannot manipulate such views risk mis-reading the data or failing altogether when the answer is hidden behind a tooltip or an axis change.

Most existing chart-centric evaluations (Masry et al., 2022; Kantharaj et al., 2022; Wu et al., 2024; Xia et al., 2024; Xu et al., 2023; Masry et al., 2025) frame the task as visual question answering (VQA) on a *single, static* bitmap. Although valuable, this setting omits two crucial aspects of real dashboards: **(1) interaction**: users (or agents) change scales, zoom, brush to reveal tooltips, or drill down; **(2) multi-view reasoning**: modern dashboards juxtapose several linked charts so that actions in one update the others, enabling richer comparisons and what-if analysis (Wang Baldonado et al., 2000; Heer & Shneiderman, 2012). Consequently, an agent that excels on static ChartVQA may still be helpless in Tableau[2], Power BI[3], or Plotly[4] dashboards—environments where answers are rarely visible in a single bitmap.

---

[1]We will make all code and data publicly available upon acceptance.

[2]https://www.tableau.com/

[3]https://www.microsoft.com/en-us/power-platform/products/power-bi

[4]https://dash.plotly.com/

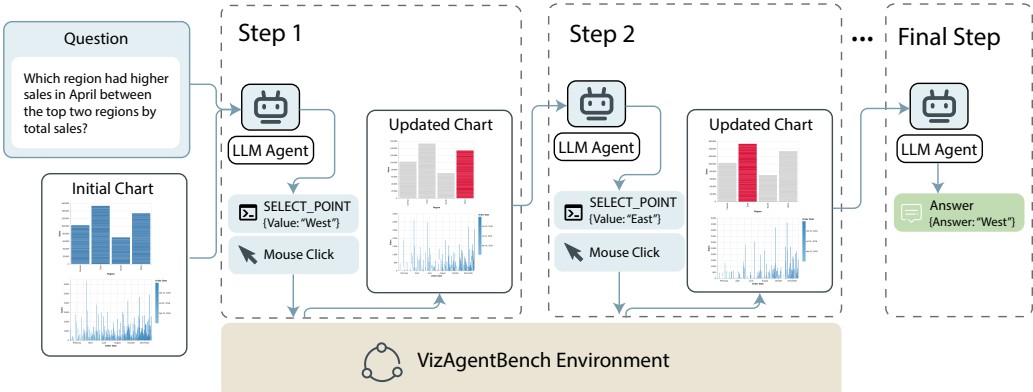

Figure 1: The workflow of VizAgentBench. (1) Initially, a screenshot of the initial visualization viewpoint and an analysis question are provided to the agent as input. (2) The agent manipulates the visualization with either mouse moves and clicks, or a command-based interface. The resulting new screenshot is provided to the agent by the environment. (3) The agent repeats generating new commands to change the visualization until an answer is reached by looking at the charts.

To close the interaction gap, in this paper, we introduce **VizAgentBench**, the first benchmark that evaluates agents on *live visualization interactions*. As illustrated in Figure 1, each task embeds an MCV dashboard in a sandboxed web page and provides the agent with a live screenshot of the current viewpoint and an operational interface (a graphical user interface or command interface). Agents must generate sequences of commands or mouse operations, perceive the updated screenshots, and finally answer an analytical question stated only in natural language. Ground-truth answers are derived from the underlying data but *not* exposed to the agent, forcing genuine visual–numerical reasoning.

To create diverse yet realistic dashboards, we begin by surveying 14 visualization research papers (Chen et al., 2022; Elias & Bezerianos, 2011; Scherr, 2008; Boukhelifa et al., 2003; North & Shneiderman, 1997) to derive a design space comprising 25 canonical chart and multi-coordinated view (MCV) templates. We then mine 10 publicly available Kaggle datasets spanning domains such as finance, healthcare, sports, and socioeconomic. Each template is parameterized using SQL queries over these datasets and rendered using d3.js. To generate question–SQL pairs aligned with the visible marks, we employ GPT-4.1 and subsequently conducted a manual inspection to ensure quality. The resulting benchmark includes 192 MCVs and the same number of questions, with guaranteed diversity across chart families, coordination types, and analytical skills including lookup, comparison, aggregation, and anomaly detection.

To summarize, our contributions are as follows: (1) We formulate the first benchmark that tests agents on interactive, multi-view visualization reasoning with a scalable generation pipeline for dashboards and question–answer pairs grounded in real-world data. (2) We release an open-source environment and API that separates perception (screenshots) from action (mouse operations/commands), facilitating rapid agent research without any proprietary renderer. (3) We conduct a comprehensive evaluation of 4 agent frameworks and 7 state-of-the-art MLLMs, revealing substantial headroom for progress.

## 2 RELATED WORK

**Agent Interaction with GUI** Recent work has moved beyond static-image understanding toward agents that can perceive a live screen and manipulate full graphical user interfaces. Vision–language–action models such as OpenAI's Computer-Using Agent (CUA)—the engine behind the Operator research preview—combine GPT-4o's visual parsing with reinforcement-learned cursor and keyboard control to execute multi-step desktop tasks (OpenAI, 2025a). In parallel, Anthropic's "computer use" mode for Claude 3.5 exposes a public-beta API that lets developers direct the model to click, scroll, and type across arbitrary applications, demonstrating competent end-to-end office workflows (Anthropic, 2024).

To evaluate these capabilities, sandboxed web environments have emerged. WebArena renders realistic e-commerce, forum, and documentation sites, turning high-level natural-language goals into fine-grained DOM interactions (Zhou et al., 2023), while BrowserGym provides a lightweight, OpenAI-Gym-style interface for scripted web tasks and leaderboarded benchmarks (Chezelles et al., 2024). Beyond HTML, ScreenAgent records real desktop sessions and trains agents with iterative plan–act–reflect loops (Niu et al., 2024), and GUI-World supplies multimodal videos, keyframes, and QA pairs spanning six common GUI scenarios (Chen et al., 2024). Very recently, the UFO 2 framework generalizes these ideas to a "desktop OS" where a HostAgent decomposes instructions across specialized AppAgents for Windows software (Zhang et al., 2025).

Despite this rapid progress, existing GUI benchmarks center on generic productivity chores—file management, form filling, simple web navigation—leaving a gap in fine-grained analytical interaction with data visualizations. Our proposed VizAgentBench fills this niche by requiring agents to combine rich visual perception, quantitative reasoning, and dynamic tool invocation within a single, continuous visualization interface.

**Benchmarks** Multimodal large language models (MLLMs) are increasingly tested on their ability to understand and generate insights from charts and data visualizations. Early synthetic benchmarks like FigureQA (Kahou et al., 2017) and DVQA (Kafle et al., 2018) provided initial testbeds, but recent years have seen more realistic and complex benchmarks. These benchmarks range from static chart question answering (QA) datasets to interactive, multi-turn dialogue and visualization-generation tasks. ChartQA (Masry et al., 2022) introduces a single-turn QA task, based on a given static chart image, with different types (bar, line, pie, etc.). OpenCQA (Kantharaj et al., 2022) adds contexts to charts (e.g., captions, articles) and requires tested models to generate answers based on both charts and contexts. SciGraphQA (Li & Tajbakhsh, 2023) collects graphs from academic preprints and creates a synthetic multiple-turn dialogue dataset with LLM. ChartInsights (Wu et al., 2024) introduces a task targeting "low-level" chart question answering, including identifying extrema, reading exact values, finding correlations, comparing categories, detecting anomalies, etc. ChartX (Xia et al., 2024) presents a dataset that has 18 diverse chart types in different domains, along with underlying data, code used to generate visualization and a text description. MMC-Benchmark (Liu et al., 2023a) introduces a large instruction-tuning dataset and a benchmark with 9 reasoning tasks. ChartBench (Xu et al., 2023) scales up the data size and includes 66.6k charts and 600k question-answer pairs. More recently, ChartQAPro (Masry et al., 2025) collects a diverse set of real-world charts paired with different types of questions.

Different from the aforementioned works, VizAgentBench is a benchmark that allows AI agents to interact with a dynamic live visualization. Our environment enables multi-step reasoning and tool calling, thus it can be an ideal testbed for MLLMs designed to power agent applications.

## 3 BACKGROUND: MULTIPLE COORDINATED VIEWS IN VISUAL ANALYTICS

Multiple Coordinated Views (MCVs) are a foundational paradigm in visual analytics that enable users to interactively explore data across several linked visualizations. Rather than relying on a single chart, MCV systems allow users to examine different aspects of the data through complementary perspectives—such as aggregations, distributions, or temporal changes—while maintaining consistent context through interaction. For example, brushing over a histogram may highlight related entries in a scatterplot, or selecting a category in a pie chart may filter a time series view to show only that subset. These cross-view interactions support core exploratory tasks like comparison, drill-down, filtering, and correlation discovery. We demonstrate an example of MCV in Figure 2.

To formalize the types of interactions used in MCV systems, Nebula (Chen et al., 2022) proposed a widely adopted taxonomy. Table 1 summarizes the seven primary interaction types that define how views respond to user actions. These interaction types are not isolated but can be composed across views to form coordination patterns. Nebula's empirical analysis of existing visual analytics (VA) systems shows that the most frequent coordination compositions include *select → filter*, *select → navigate*, and *navigate → navigate*. Such patterns allow MCV systems to support common EDA operations such as narrowing down data, refining focus, and comparing linked subsets.

We adopt these patterns to guide the design of our benchmark's coordination logic. Specifically, we define each MCV template as a set of coordinated views, linked by a series of one-way coordination

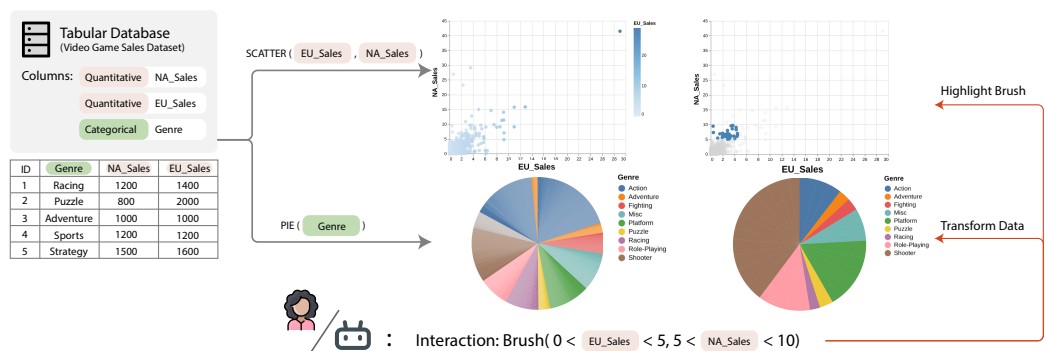

Figure 2: An example of Multiple Coordinated Views (MCVs). This visualization includes two charts: (1) A scatter plot that illustrates the relationship between a video game's sales in Europe (EU_Sales) and North America (NA_Sales). (2) A pie chart that shows the genres of the selected data points in the scatter plot. A human or AI Agent can use the brush tool to select points from the scatter plot, to take a closer look at some data and analyze, and answer questions like "Which genre is popular in NA but unpopular in EU?"

Table 1: Coordination taxonomy in coordinated visualizations (adapted from Chen et al. (2022)).

| Coordination Type | Description |
|---|---|
| **Select** | Specify a subset of data items through actions like clicking or brushing. |
| **Filter** | Limit the visible data points based on inclusion criteria. |
| **Navigate** | Modify the viewport, such as zooming or panning. |
| **Encode** | Change the visual encoding channels (e.g., color, size). |
| **Reconfigure** | Rearrange the layout or structure of the visualization, such as sorting or switching axes. |
| **Set** | Adjust parameters or controls, such as sliders and toggles. |
| **Append** | Add new data without replacing existing content, common in streaming or real-time settings. |

patterns sampled from the Nebula-derived set. For example, a histogram and a bar chart can be linked via a *select → filter* pattern, where brushing a time range in the histogram filters the bar chart by date. These design choices are grounded in real-world practice and reflect well-established principles in visualization systems like Snap-Together Visualization (North & Shneiderman, 2000) and the coordination taxonomy by Wang Baldonado et al. (2000) summarized in Table 1. In the following section, we will describe the design and generation of the MCVs.

## 4 BENCHMARK DESIGN

Our benchmark evaluates LLM agents on interactive reasoning tasks over coordinated visualizations. It includes a template-driven chart generator, a question-answer creation pipeline based on LLMs, and an interactive chart execution environment. Additional implementation details and source datasets for MCV generation can be found in Appendix B.

### 4.1 TEMPLATE CONSTRUCTION VIA COORDINATION LOGIC

Each benchmark instance consists of 2-4 coordinated charts linked through multiple defined interactions. We support five fundamental chart types: bar chart, pie chart, line chart, scatter plot, and histogram. These cover a broad range of analytical idioms including distribution, trend, and categorical comparison. For each chart, we assign data fields to encoding channels based on the semantic type of the column, using the mapping shown in Table 2.

Table 2: Supported chart types and their encoding strategy based on field types.

| Chart Type | X Encoding | Y Encoding / Mark |
|---|---|---|
| Bar Chart | Categorical | Aggregated quantitative (height) |
| Pie Chart | Categorical | Aggregated quantitative (angle) |
| Line Chart | Temporal | Quantitative (line) |
| Scatter Plot | Quantitative | Quantitative (points) |
| Histogram | Binned quantitative | Count or sum (height) |

Table 3: Interaction modalities supported in the benchmark. Each mouse interaction has an equivalent command that produces identical effects. "✓" indicates which chart types support each interaction.

| Mouse Interaction | Command | Function | Applicable charts | | | | |
|---|---|---|---|---|---|---|---|
| | | | Bar | Line | Pie | Hist. | Scatter |
| Click on element | `select_point` | Select specific data point for filtering or highlighting | ✓ | ✓ | ✓ | ✓ | ✓ |
| Click and drag | `brush_range` | Define a range of values for filtering | ✓ | ✓ | | ✓ | ✓ |
| Click on legend | `select_legend` | Filter by categorical value | ✓ | ✓ | ✓ | | ✓ |
| Pan/zoom gestures | `navigate` | Change the visible range of the visualization | | ✓ | | | ✓ |
| Mouse move | `move_cursor` | Update the line chart's time state | | ✓ | | | |

To create coordinated views, we define a coordination function $\Phi(c_s, c_t, i, f)$ that links a source chart $c_s$ to one or more target charts $c_t$ via coordination $i$ on a shared field $f$. Multiple coordination functions can be defined between different pairs of views in the same MCV. For instance, a coordination $\Phi(\text{Histogram}, \text{Line}, select \rightarrow filter, \text{Order Date})$ allows brushing a time range in the histogram to filter the line chart, while another coordination might link a bar chart selection to a scatter plot. This formulation enables instantiation of realistic coordination patterns, such as $select \rightarrow filter$, $select \rightarrow navigate$, and $select \rightarrow reconfigure$ across multiple views.

The benchmark templates are parameterized combinations of multiple chart types (2-4 views) and coordination patterns. Each template is defined as a function following a standardized interface, where multiple D3.js views (e.g., line, bar, scatter) are composed with specific coordination logic. Templates are manually authored based on our custom MCV framework (described in Appendix C) and view coordinations (see Table 1), ensuring clarity and consistency across configurations. At runtime, templates are instantiated by sampling appropriate field types from the dataset schema and applying the corresponding rendering function. This design allows us to systematically scale the generation of diverse and grounded MCV chart configurations.

We manually craft 25 unique templates. These templates cover a wide spectrum of view combinations and coordinations. The templates vary in complexity, from simpler two-view configurations to more complex four-view MCVs with multiple coordination relationships. This modular structure supports both interactive evaluation and synthetic benchmark generation.

### 4.2 LLM-GUIDED QUESTION AND ANSWER GENERATION

To generate benchmark tasks, we use LLMs to synthesize natural language questions, specify the chart configuration, and produce SQL expressions for computing ground-truth answers. Each instance consists of a triplet $(q, c, \texttt{SQL})$, where $q$ is a question referring to the visualization $c$ and $\texttt{SQL}$ yields the correct answer.

Following recent trends in benchmark construction using LLMs (Wang et al., 2022; Liu et al., 2023b; Luo et al., 2023), we use GPT-4.1 to generate natural language questions, configure chart templates, and synthesize SQL programs for computing ground-truth answers. For example, given a line-pie

Table 4: Statistics of chart usage across different MCV configurations in the benchmark. The table shows the distribution of chart types across 2-view, 3-view, and 4-view coordinated visualizations.

| Chart Type | 2-View MCVs | 3-View MCVs | 4-View MCVs |
|---|---|---|---|
| Bar | 36 | 54 | 80 |
| Line | 20 | 42 | 54 |
| Pie | 20 | 42 | 60 |
| Histogram | 20 | 30 | 50 |
| Scatter | 20 | 60 | 60 |
| **Total** | 58 | 58 | 76 |

coordinated chart, the model may be asked to generate a comparison question such as *"Which sub-category has the largest sales peak in the second half of the year?"* and compute the answer using an aggregation-and-filtering SQL expression.

All generated questions and ground-truth answers are manually inspected by a group of students majored in data science for quality. Importantly, the evaluated LLM agent does not have access to the SQL or data; it must derive the answer by observing and interacting with the visual interface.

### 4.3 INTERACTIVE BENCHMARK ENVIRONMENT

To support visual reasoning and interaction, we implement an interactive benchmark environment using D3.js and our custom MCV framework (detailed in Appendix B.1). Our benchmark supports two complementary interaction modalities: traditional agent-specific command-based interaction and human-like mouse interaction. Importantly, both interaction types provide consistent affordances, ensuring no features are exclusive to either modality.

Each task begins with an initial chart rendered via D3.js and saved to a PNG file. The image is provided to the LLM agent, which can interact with the visualization through either command-based or mouse-based interactions. As shown in Table 3, there is a direct one-to-one mapping between command types and their equivalent mouse interactions. For example, a `select_point` command corresponds to clicking on a visual element with a mouse, while a `brush_range` command corresponds to clicking and dragging to create a range selection.

## 5 EXPERIMENTS

### 5.1 EXPERIMENTAL SETTINGS

**Dataset Statistics**   VizAgentBench has 10 databases, 25 manually crafted visualization templates and 192 questions. The chart composition of the 192 questions are detailed in Table 4. As shown in the table, our benchmark provides a balanced distribution across different chart types, with each type appearing in both source (first) and target (second) positions in the coordinated views.

**Evaluation Metrics**   We assess each model using the Pass@k metric: Pass@1 measures first-attempt accuracy, while Pass@3 captures success within three attempts.

**Evaluated LLMs**   We evaluate models from five major AI labs: OpenAI (o3, o4-mini, GPT-5), Anthropic (Claude 3.7 Sonnet with and without thinking mode), Google (Gemini-2.5-Pro), Alibaba (Qwen-2.5-VL 75B), and Meta (LLaMA-4-Maverick). These models represent the current state-of-the-art in multimodal capabilities, with varying architectures and training approaches. All models use the default or officially recommended temperature settings.

**Agent Frameworks**   We test four distinct agent frameworks: **(1) OpenAI-Agents** (OpenAI, 2025b): A framework developed by OpenAI that implements structured planning and execution with state management capabilities for complex tasks. **(2) OpenManus** (Liang et al., 2024) is an open-source framework for building and benchmarking multimodal LLM agents. It provides standardized agent

Table 5: Comparison of agent frameworks used in our evaluation. All frameworks support command-based interactions, while only OpenAI Agents SDK and OpenManus support mouse interactions. Memory components and native tool support vary across frameworks. *ReAct's tool support was implemented by us for this benchmark.

| Framework | Command-based Interaction | Mouse Interaction | Memory Components | Tool Support |
|---|---|---|---|---|
| OpenAI Agents SDK (OpenAI, 2025b) | ✓ | ✓ | ✓ | ✓ |
| OpenManus Liang et al. (2024) | ✓ | ✓ | ✓ | ✓ |
| AutoGen (Wu et al., 2023) | ✓ | | | ✓ |
| ReAct (Yao et al., 2023) | ✓ | | | ✓* |

interfaces, plan–act–reflect execution, and support for evaluation on complex interactive tasks. **(3) ReAct** (Yao et al., 2023): A conceptual framework that combines reasoning and acting by interleaving three key components: (a) verbal reasoning traces that explicitly document the agent's thought process, (b) action planning based on these traces, and (c) observation of action outcomes to inform subsequent reasoning steps. **(4) AutoGen** (Wu et al., 2023): A multi-agent framework developed by Microsoft Research that supports conversational problem-solving through structured agent communication and memory mechanisms for tracking interaction history. We compare the agent frameworks in Table 5.

Each model-framework combination is evaluated on the same set of tasks, with a maximum of 10 turns of interaction allowed per question. This comprehensive evaluation allows us to assess both the capabilities of the underlying models and the effectiveness of different agent architectures in visual analytics contexts.

## 5.2 BENCHMARK RESULTS

**Impact of MLLMs** We evaluate state-of-the-art MLLMs on VizAgentBench across multiple agent frameworks, with results shown in Table 6. Our evaluation reveals significant performance differences across both models and agent frameworks. GPT-5 with OpenAI-Agents achieves the highest Pass@1 accuracy at 44.2%, followed by OpenAI o3 with OpenAI-Agents at 42.1%. The OpenAI models consistently outperform other families, with o3 and o4-mini showing strong results across all agent frameworks. When examining Pass@3 metrics, the relative ranking of models remains largely consistent between Pass@1 and Pass@3, the performance gap narrows, suggesting that weaker models can partially compensate through multiple attempts.

Claude 3.7 Sonnet with thinking mode enabled (37.0% with OpenManus) outperforms its standard version (32.5% with AutoGen), confirming the value of explicit reasoning for visual analytics tasks. Qwen-2.5-VL 75B demonstrates competitive performance (32.6% with OpenManus) for an open-source model, likely due to its specific training on chart data and optimization for visual agent tasks as noted in Bai et al. (2025). Gemini-2.5-Pro (30.4% with OpenManus) and LLaMA-4-Maverick (30.5% with OpenManus) show comparable performance but lag behind the leading models by a significant margin.

**Impact of Agent Frameworks** The choice of agent framework significantly affects performance across all models. OpenManus consistently delivers high results for each model family, with OpenAI-Agents showing particularly effective performance with OpenAI models. As a baseline framework, ReAct generally shows lower performance compared to other frameworks. The performance gap between frameworks (up to 7.3 percentage points for LLaMA-4-Maverick) highlights the importance of agent architecture in multimodal reasoning tasks involving interactive visualizations.

## 5.3 CASE STUDY: MULTI-HOP VISUALIZATION INTERACTIONS FOR AUTOMOBILE ANALYSIS

To showcase our benchmark's capability to evaluate LLM agents' reasoning and tool-calling abilities, we analyze a representative task from the automobile dataset. The question asks: *"For the origin with the highest horsepower, what is the least common cylinder number? For the automobiles with this cylinder number, is the horsepower above or below average?"* Shown in Figure 3, the MCV consists

Table 6: Benchmark accuracy on visually grounded MCV tasks. Pass@k is the proportion (%) of tasks solved within the top-$k$ sampled plans/answers.

| LLM model | Agent framework | Pass@1 | Pass@3 |
|---|---|---|---|
| *OpenAI (o3 / o4-mini / GPT-5)* | | | |
| OpenAI o3 | OpenAI-Agents | 42.1 | **63.7** |
| OpenAI o3 | OpenManus | 41.5 | 61.3 |
| OpenAI o3 | ReAct | 38.4 | 62.2 |
| OpenAI o4-mini | OpenAI-Agents | 39.5 | 60.1 |
| OpenAI o4-mini | OpenManus | 39.1 | 61.8 |
| OpenAI o4-mini | ReAct | 34.8 | 52.0 |
| GPT-5 | OpenAI-Agents | **44.2** | 62.4 |
| GPT-5 | OpenManus | 41.8 | 58.9 |
| GPT-5 | ReAct | 40.4 | 60.2 |
| *Anthropic (Claude 3.7)* | | | |
| Claude-3.7-Sonnet Thinking-Mode | OpenManus | 37.0 | 60.2 |
| Claude-3.7-Sonnet Thinking-Mode | AutoGen | 36.8 | 58.4 |
| Claude-3.7-Sonnet Thinking-Mode | ReAct | 35.2 | 58.3 |
| Claude-3.7-Sonnet | AutoGen | 32.5 | 51.2 |
| Claude-3.7-Sonnet | ReAct | 31.8 | 53.0 |
| *Google (Gemini)* | | | |
| Gemini-2.5-Pro | OpenManus | 30.4 | 45.2 |
| Gemini-2.5-Pro | AutoGen | 29.2 | 43.8 |
| Gemini-2.5-Pro | ReAct | 28.5 | 40.0 |
| *Alibaba (Qwen)* | | | |
| Qwen-2.5-VL 75B | OpenManus | 32.6 | 42.3 |
| Qwen-2.5-VL 75B | ReAct | 30.2 | 40.6 |
| *Meta (LLaMA)* | | | |
| LLaMA-4-Maverick | OpenManus | 30.5 | 40.0 |
| LLaMA-4-Maverick | ReAct | 27.7 | 31.2 |

of a bar chart showing horsepower by origin, a pie chart displaying cylinder number distribution, another pie chart showing different origins, and a histogram visualizing the horsepower distribution.

The gold standard solution requires a multi-step interaction sequence: first, identify the origin with highest horsepower from the bar chart (USA); second, select this origin to filter the cylinder distribution pie chart; third, identify the least common cylinder number (3 cylinders); and finally, select this cylinder number to view its horsepower distribution in the histogram and compare it with the overall average.

Our analysis reveals three critical challenges that prevent LLM agents from completing this task successfully: **(1) Visual recognition errors significantly impact performance.** Gemini-2.5-Pro ReAct agents frequently misidentified the highest horsepower origin or failed to correctly read the filtered pie chart values, leading to incorrect cylinder identification. These errors compound through the interaction chain, resulting in completely incorrect answers despite proper execution of the interaction mechanics. **(2) Maintaining visual memory across sequential states proves challenging.** Since the task requires comparing the filtered histogram (showing only 3-cylinder cars) with the overall distribution, agents must remember the previous state. GPT-o3 ReAct agents often failed to maintain this memory, making comparison judgments based solely on the current view without reference to the previous distribution. **(3) Multi-turn context management presents significant difficulties.** Qwen ReAct agents initially identified the correct origin with highest horsepower but lost context during exploration. When examining the pie chart after filtering, they repeatedly focused on the original bar chart values rather than the newly filtered pie chart, demonstrating an inability to maintain interaction context across turns.

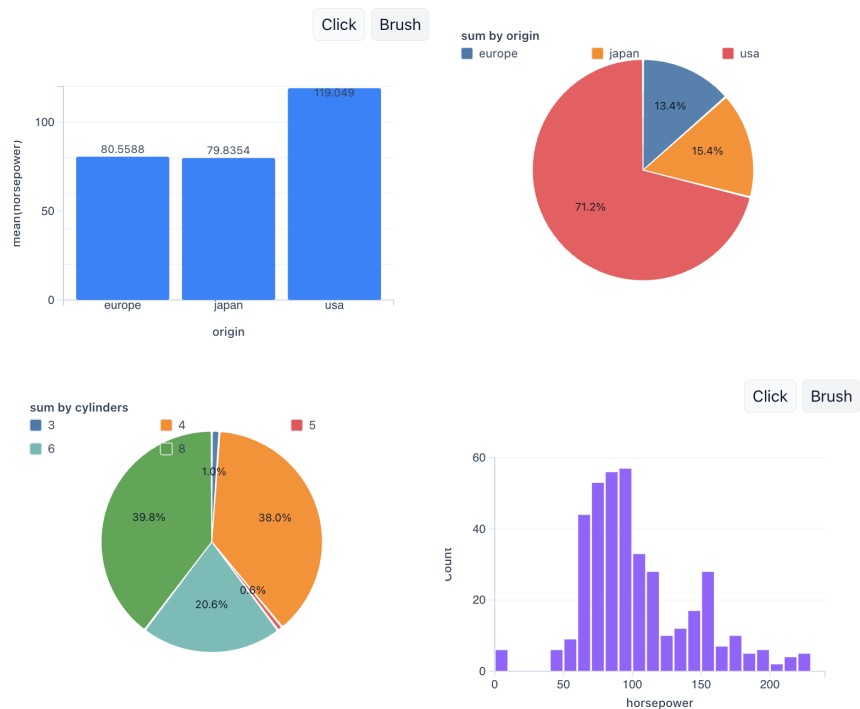

Figure 3: Example of a multiple coordinated view (MCV) from the automobile dataset showing horsepower, origin and number of cylinders information across four linked views.

This case study illustrates that successful completion of complex MCV tasks requires a combination of accurate visual perception, robust memory of previous states, and consistent context maintenance across multiple interaction turns—capabilities that vary significantly across current LLM agent implementations. We observed that the current LLM agent frameworks may partially cover the required capabilities, leading to the pitfall of the task. Furthermore, even though the LLM agents may have three capabilities, the agent is unsteady in the multiple runs of the task, reflected in our Pass@3 score. We observed that for the same LLM agent, the error may happen in the different stages, indicating the long reasoning chain may destabilize the LLM agent output, limiting its effectiveness in the realistic tasks.

## 6 CONCLUSION AND FUTURE WORK

We introduced VizAgentBench, the first benchmark evaluating multimodal agents on interactive, multi-view visualization reasoning. Our evaluation of state-of-the-art MLLMs across different agent frameworks reveals significant challenges in this domain. Even top-performing models (GPT-5 with OpenAI-Agents at 44.2% Pass@1) struggle with complex visual analytics tasks, particularly with maintaining visual memory across sequential interactions and executing multi-hop reasoning chains. The substantial performance gap between frameworks (with OpenManus and OpenAI-Agents consistently outperforming others) highlights the critical importance of agent architecture in multimodal reasoning.

Our analysis shows that models with explicit reasoning capabilities demonstrate enhanced performance on complex visual analytics tasks, suggesting that bridging reasoning and multimodality is a promising research direction. For future work, we plan to diversify visualization patterns for broader business intelligence use cases, explore specialized agent architectures for visual analytics, and develop metrics that better capture interaction quality. By providing an open-source environment that separates perception from action, VizAgentBench establishes a foundation for advancing MLLMs' capabilities in interactive visualization reasoning, paving the way toward more collaborative AI systems that can support data-driven decision making across diverse real-world domains.

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

## A USE OF LLMs IN PAPER WRITING

We acknowledge the use of large language models (LLMs) in the preparation of this manuscript. Specifically, we used LLMs to proofread and revise the text, ensuring clarity, coherence, and adherence to academic writing standards. Additionally, LLMs were employed to format the LaTeX table in Table 6, helping to structure the complex benchmark results in a readable format. The use of LLMs was limited to editorial assistance and formatting tasks, while all research design, experiments, analysis, and scientific conclusions were conducted and drawn by the authors.

## B IMPLEMENTATION DETAILS AND SOURCE DATASETS

### B.1 IMPLEMENTATION DETAILS

**System Stack** The benchmark system is implemented in Python and designed with modularity to support reproducibility, extensibility, and compatibility with a range of multimodal language models. The visual analytics interface is based on D3.js and our custom MCV framework, with charts constructed with explicit interaction logic through an affordance abstraction. Each chart declares a set of supported interactions (e.g., `select_legend`, `brush_range`) that can be triggered via structured JSON commands issued by the agent. This design allows precise control over interaction semantics while keeping the chart specifications declarative and immutable.

**Agent-Environment Interaction** Agents interact with the system through an iterative loop. At each step, the model receives the current chart image and a task prompt, and responds with either a structured interaction command, mouse-based interaction, or final answer. Vision-capable models interpret the chart visually, while others may receive auxiliary descriptions or structured metadata. Both command-based and mouse-based interactions are translated to the same underlying chart operations, ensuring consistent behavior regardless of interaction method. The interaction responses are applied to update the chart state, and the process continues until the model chooses to submit an answer.

**Prompt Design** Prompts are defined using versioned templates with structured metadata and filled dynamically with task-specific variables. The prompting system supports consistent and configurable task formulation across models. For vision-based interaction, chart images are rendered and served to the agent; for non-vision models, descriptions can be injected using auxiliary tools.

**Question Generation** Natural language questions are generated using a scripted prompting pipeline applied to GPT-4.1. For each chart configuration, questions are designed to test a specific analytical pattern such as trend identification, group comparison, or anomaly detection. These questions are paired with metadata that defines expected chart fields and interaction types.

**Evaluation** Evaluation of final answers is conducted using two complementary strategies. When a ground-truth answer is known, exact or soft string matching is used. For data-derived tasks, correctness is computed programmatically using logic based on the original data, supporting both scalar and range-based answers. These metrics allow us to evaluate both factual correctness and alignment with the underlying chart semantics.

We use a variety of tabular datasets, with an emphasis on those that exhibit diverse data types and realistic schemas. The Superstore dataset is the primary benchmark source due to its rich mix of temporal, categorical, and quantitative fields. Additional datasets are selected from public sources, including domains such as finance, operations, and marketing. All datasets are preprocessed to ensure semantic consistency and support a broad range of visual encoding strategies.

### B.2 SOURCE DATASETS

We use 10 different source datasets available online to generate MCVs, as the Automobile Dataset[5], Netflix Movies and TV Shows[6], Bundesliga Player Stats[7], Retail Sales Data[8], Video Game Sales[9], European Ski Resorts[10], Health Care[11], London Daily Weather[12], Superstore Sales[13], and the Sample Sales dataset[14].

## C  MCV TEMPLATES

Each template specifies (i) *Composition* (which views it includes), (ii) *Roles*—typed placeholders such as `category_column1` or `numeric_column1` that are bound to dataset columns at instantiation—and (iii) *Coordination*, which defines how interactions in one view affect the others (e.g., *select → filter*).

### C.1  TWO-VIEW TEMPLATES

#### C.1.1  `tpl-bar-line-coord` — BAR + LINE

**Composition:** $1 \times$ Bar, $1 \times$ Line
**Roles:**

- `category_column1`: categorical (required)
- `temporal_column1`: temporal (required)
- `numeric_column1`: numeric (required; Bar value)
- `numeric_column2`: numeric (required; Line value)

**Coordination:** Clicking a bar category filters the Line to that category and highlights the selected category on the Bar.

#### C.1.2  `tpl-bar-scatter-coord` — BAR + SCATTER

**Composition:** $1 \times$ Bar, $1 \times$ Scatter
**Roles:**

- `category_column1`: categorical (required)
- `numeric_column1`: numeric (required; Bar value, Scatter X)
- `numeric_column2`: numeric (required; Scatter Y)

**Coordination:** Clicking a bar category filters the Scatter to that category; Scatter interactions can optionally control highlight/brush states on the Bar.

#### C.1.3  `tpl-bar-pie-coord` — BAR + PIE

**Composition:** $1 \times$ Bar, $1 \times$ Pie
**Roles:**

- `category_column1`: categorical (required)
- `numeric_column1`: numeric (required)

---

[5]https://www.kaggle.com/datasets/toramky/automobile-dataset

[6]https://www.kaggle.com/datasets/shivamb/netflix-shows

[7]https://www.kaggle.com/datasets/yoanbernabeu/bundesliga-player-stats-20232024

[8]https://www.kaggle.com/datasets/abdullah0a/retail-sales-data-with-seasonal-trends-and-marketing

[9]https://www.kaggle.com/datasets/gregorut/videogamesales

[10]https://www.kaggle.com/datasets/jorisgonen/european-ski-resorts

[11]https://www.kaggle.com/datasets/prasad22/healthcare-dataset

[12]https://www.kaggle.com/datasets/zusmani/london-daily-weather-data

[13]https://www.kaggle.com/datasets/rohitsahoo/sales-forecasting

[14]https://www.kaggle.com/datasets/kyanyoga/sample-sales-data

**Coordination:** Clicking bar or pie categories updates a shared highlight across both views.

### C.1.4 `tpl-bar-histogram-coord` — BAR + HISTOGRAM

**Composition:** $1 \times$ Bar, $1 \times$ Histogram
**Roles:**

- `category_column1`: categorical (required)
- `numeric_column1`: numeric (required; both views)

**Coordination:** Clicking a bar category filters the Histogram to that category and highlights the Bar.

### C.1.5 `tpl-histogram-pie-coord` — HISTOGRAM + PIE

**Composition:** $1 \times$ Histogram, $1 \times$ Pie
**Roles:**

- `numeric_column1`: numeric (required; Hist + Pie value)
- `category_column1`: categorical (required; Pie category)

**Coordination:** Brushing a numeric range in the Histogram filters the Pie.

### C.1.6 `tpl-line-scatter-coord` — LINE + SCATTER

**Composition:** $1 \times$ Line, $1 \times$ Scatter
**Roles:**

- `temporal_column1`: temporal (required; Line X)
- `numeric_column1`: numeric (required; Line Y, Scatter X)
- `numeric_column2`: numeric (required; Scatter Y)

**Coordination:** Brushing a temporal range in the Line highlights corresponding points in the Scatter.

### C.1.7 `tpl-line-histogram-coord` — LINE + HISTOGRAM

**Composition:** $1 \times$ Line, $1 \times$ Histogram
**Roles:**

- `temporal_column1`: temporal (required; Line X)
- `numeric_column1`: numeric (required; Line Y, Histogram field)

**Coordination:** Brushing a numeric range in the Histogram filters the Line to values in that range.

### C.1.8 `tpl-line-pie-coord` — LINE + PIE

**Composition:** $1 \times$ Line, $1 \times$ Pie
**Roles:**

- `temporal_column1`: temporal (required; Line X)
- `numeric_column1`: numeric (required; Line Y, Pie value)
- `category_column1`: categorical (required; Pie category)

**Coordination:** Selecting a slice in the Pie highlights the corresponding category trend in the Line.

### C.1.9 `tpl-scatter-histogram-coord` — SCATTER + HISTOGRAM

**Composition:** $1 \times$ Scatter, $1 \times$ Histogram
**Roles:**

- `numeric_column1`: numeric (required; Scatter X, Histogram field)
- `numeric_column2`: numeric (required; Scatter Y)

**Coordination:** Brushing a numeric range in the Histogram filters points in the Scatter.

### C.1.10 `tpl-bar-bar-coord` — BAR + BAR

**Composition:** $1 \times$ Bar (A), $1 \times$ Bar (B)
**Roles:**

- `category_column1`: categorical (required; shared category)
- `numeric_column1`: numeric (required; Bar A value)
- `numeric_column2`: numeric (required; Bar B value)

**Coordination:** Clicking a category in either Bar highlights the same category in the other view; optional cross-filtering can restrict to the selected category.

### C.1.11 `tpl-line-line-coord` — LINE + LINE

**Composition:** $1 \times$ Line (A), $1 \times$ Line (B)
**Roles:**

- `temporal_column1`: temporal (required; shared time axis)
- `numeric_column1`: numeric (required; Line A Y)
- `numeric_column2`: numeric (required; Line B Y)

**Coordination:** Brushing a temporal range in either Line synchronizes the range in both views; hover highlights the corresponding point across Lines.

### C.1.12 `tpl-scatter-scatter-coord` — SCATTER + SCATTER

**Composition:** $1 \times$ Scatter (A), $1 \times$ Scatter (B)
**Roles:**

- `numeric_column1`: numeric (required; Scatter A X)
- `numeric_column2`: numeric (required; Scatter A Y)
- `numeric_column3`: numeric (required; Scatter B X)
- `numeric_column4`: numeric (required; Scatter B Y)
- `category_column1`: categorical (optional; shared color/group)

**Coordination:** Lasso/brush selection in one Scatter highlights (and optionally filters) the corresponding points in the other via shared row IDs or grouping.

## C.2 THREE-VIEW TEMPLATES

### C.2.1 `tpl-bar-line-scatter` — BAR + LINE + SCATTER

**Composition:** $1 \times$ Bar, $1 \times$ Line, $1 \times$ Scatter
**Roles:**

- `category_column1`: categorical (required; Bar category, optional Scatter color)
- `temporal_column1`: temporal (required; Line X)
- `numeric_column1`: numeric (required; Bar value, Scatter X)
- `numeric_column2`: numeric (required; Line Y, Scatter Y)

**Coordination:** Clicking a Bar category filters both the Line (to that category) and the Scatter; hover/selection highlights the chosen category across views.

### C.2.2 `tpl-bar-scatter-histogram` — BAR + SCATTER + HISTOGRAM

**Composition:** $1 \times$ Bar, $1 \times$ Scatter, $1 \times$ Histogram
**Roles:**

- `category_column1`: categorical (required; Bar category)
- `numeric_column1`: numeric (required; Bar value, Scatter X, Histogram field)
- `numeric_column2`: numeric (required; Scatter Y)

**Coordination:** Clicking a Bar category filters the Scatter and the Histogram to the selected category; brushing a numeric range in the Histogram further filters points in the Scatter (composing with the Bar selection).

### C.2.3 `tpl-bar-pie-scatter` — BAR + PIE + SCATTER

**Composition:** $1 \times$ Bar, $1 \times$ Pie, $1 \times$ Scatter
**Roles:**

- `category_column1`: categorical (required; Bar & Pie category, optional Scatter color)
- `numeric_column1`: numeric (required; Bar/Pie value, Scatter X)
- `numeric_column2`: numeric (required; Scatter Y)

**Coordination:** Clicking a Bar category or a Pie slice updates a shared category selection that filters the Scatter and highlights the selection in both Bar and Pie.

### C.2.4 `tpl-bar-2scatter` — BAR + 2 SCATTER

**Composition:** $1 \times$ Bar, $2 \times$ Scatter (A & B)
**Roles:**

- `category_column1`: categorical (required; Bar category, optional color/group)
- `numeric_column1`: numeric (required; Bar value, Scatter A X)
- `numeric_column2`: numeric (required; Scatter A Y)
- `numeric_column3`: numeric (required; Scatter B X)
- `numeric_column4`: numeric (required; Scatter B Y)

**Coordination:** Clicking a Bar category filters both Scatter views; lasso/brush selection in either Scatter highlights (and optionally filters) the corresponding subset in the other Scatter and the Bar.

### C.2.5 `tpl-scatter-line-pie` — SCATTER + LINE + PIE

**Composition:** $1 \times$ Scatter, $1 \times$ Line, $1 \times$ Pie
**Roles:**

- `category_column1`: categorical (required; Pie category, optional Scatter color)
- `temporal_column1`: temporal (required; Line X)
- `numeric_column1`: numeric (required; Line Y, Scatter X, Pie value)
- `numeric_column2`: numeric (required; Scatter Y)

**Coordination:** Selecting a Pie slice updates a shared category selection that highlights corresponding points in the Scatter and filters the Line to that category when applicable.

### C.2.6 `tpl-scatter-histogram-line` — SCATTER + HISTOGRAM + LINE

**Composition:** $1 \times$ Scatter, $1 \times$ Histogram, $1 \times$ Line
**Roles:**

- `temporal_column1`: temporal (required; Line X)
- `numeric_column1`: numeric (required; Line Y, Scatter X, Histogram field)
- `numeric_column2`: numeric (required; Scatter Y)

**Coordination:** Brushing a numeric range in the Histogram filters both the Scatter and the Line to records whose `numeric_column1` falls within the selected range; hover/selection in Scatter can highlight the corresponding distribution in the Histogram.

## C.3   FOUR-VIEW TEMPLATES

### C.3.1   `tpl-bar-pie-hist-scatter` — BAR + PIE + HISTOGRAM + SCATTER

**Composition:** $1 \times$ Bar, $1 \times$ Pie, $1 \times$ Histogram, $1 \times$ Scatter
**Roles:**

- `category_column1`: categorical (required)
- `numeric_column1`: numeric (required; Bar/Pie/Hist/Scatter X)
- `numeric_column2`: numeric (required; Scatter Y)

**Coordination:** Selecting a category in Bar/Pie filters the Histogram and sets highlights across all views.

### C.3.2   `tpl-bar-2pie-hist` — BAR + PIE + PIE + HISTOGRAM

**Composition:** $1 \times$ Bar, $2 \times$ Pie, $1 \times$ Histogram
**Roles:**

- `category_column1`: categorical (required; Bar + Pie A)
- `category_column2`: categorical (optional; Pie B)
- `numeric_column1`: numeric (required; Bar/Pie/Hist)

**Coordination:** Clicking a Bar category filters Pie A and Histogram, highlighting the Bar selection.

### C.3.3   `tpl-bar-scatter-2hist` — BAR + SCATTER + 2 HISTOGRAMS

**Composition:** $1 \times$ Bar, $1 \times$ Scatter, $2 \times$ Histogram
**Roles:**

- `category_column1`: categorical (required)
- `numeric_column1`: numeric (required; Bar/Scatter X/Hist-1)
- `numeric_column2`: numeric (required; Scatter Y/Hist-2)

**Coordination:** Clicking a Bar category filters the Scatter and both Histograms, highlighting the Bar.

### C.3.4   `tpl-line-line-bar-scatter` — 2 LINES + BAR + SCATTER

**Composition:** $2 \times$ Line, $1 \times$ Bar, $1 \times$ Scatter
**Roles:**

- `temporal_column1`: temporal (required)
- `numeric_column1`: numeric (required; Line A, Bar, Scatter X)
- `numeric_column2`: numeric (required; Line B, Scatter Y)
- `category_column1`: categorical (required; Bar + Scatter color)

**Coordination:** Clicking a Bar category filters both Lines and the Scatter, highlighting the Bar.

### C.3.5   `tpl-bar-line-scatter-histogram` — BAR + LINE + SCATTER + HISTOGRAM

**Composition:** $1 \times$ Bar, $1 \times$ Line, $1 \times$ Scatter, $1 \times$ Histogram
**Roles:**

- `category_column1`: categorical (required; Bar category, optional Scatter color)

- `temporal_column1`: temporal (required; Line X)
- `numeric_column1`: numeric (required; Bar value, Line Y, Scatter X, Histogram field)
- `numeric_column2`: numeric (required; Scatter Y)

**Coordination:** Clicking a Bar category filters the Line, Scatter, and Histogram to that category; brushing a numeric range in the Histogram further filters Line and Scatter (composing with the category selection); hover/selection highlights the chosen subset across all views.

### C.3.6   `tpl-bar-line-pie-scatter` — BAR + LINE + PIE + SCATTER

**Composition:** $1 \times$ Bar, $1 \times$ Line, $1 \times$ Pie, $1 \times$ Scatter
**Roles:**

- `category_column1`: categorical (required; Bar & Pie category, optional Scatter color)
- `temporal_column1`: temporal (required; Line X)
- `numeric_column1`: numeric (required; Bar/Pie value, Line Y, Scatter X)
- `numeric_column2`: numeric (required; Scatter Y)

**Coordination:** Clicking a Bar category or selecting a Pie slice updates a shared category selection that filters the Line and Scatter and highlights the selection in both Bar and Pie; hover/selection in Scatter can emphasize the corresponding category across views.

### C.3.7   `tpl-pie-scatter-2hist` — PIE + SCATTER + 2 HISTOGRAMS

**Composition:** $1 \times$ Pie, $1 \times$ Scatter, $2 \times$ Histogram (X & Y distributions)
**Roles:**

- `category_column1`: categorical (required; Pie category, optional Scatter color)
- `numeric_column1`: numeric (required; Scatter X, Histogram-X field)
- `numeric_column2`: numeric (required; Scatter Y, Histogram-Y field)

**Coordination:** Selecting a Pie slice filters and highlights corresponding points in the Scatter and updates both Histograms; brushing in either Histogram filters the Scatter by the selected numeric range and updates the Pie proportions accordingly (composition of category and numeric filters).

