# OpenReview forum: "VizAgentBench: Benchmarking Multimodal Agent Reasoning on Coordinated Multi-View Visual Analytics Tasks"
_ICLR.cc/2026/Conference — Submitted to ICLR 2026_

### Official Review · Reviewer_aEAe · 2025-10-27

**Soundness:** 2
**Presentation:** 2
**Contribution:** 3
**Rating:** 6
**Confidence:** 3

**Summary:**

This paper introduces VizAgentBench, the first benchmark designed to evaluate multimodal large language model (MLLM) agents on interactive, multi-view data visualization tasks. It spans 10 Kaggle datasets across diverse domains, containing 192 datasets. Evaluation across 4 agent frameworks and 7 MLLMs shows that current systems achieve only about 40% accuracy, indicating significant room for improvement.

**Strengths:**

1. Novel Problem Setting. Moves beyond static chart understanding to dynamic, interactive reasoning, aligning better with real-world analytics workflows.
2. Clear Separation of Perception and Action. The API cleanly decouples screenshot understanding from action execution, promoting modular evaluation of perception vs. planning.
3. Reproducibility and Openness. Full release of data and code lowers the entry barrier for future agent research.

**Weaknesses:**

1. Evaluation Ambiguity. It’s not entirely clear how success metrics is calculated. Also, it would be helpful to show the horizon of the task, and human performance to clarify the gap.
2. Lack of in-depth analysis. There is not a clear analysis of different error modes.
3. Extensibility. It is unclear whether the framework could support tasks beyond QA with more intensive interaction like data update or chart creation.

**Questions:**

1. Can you provide a detailed analysis of error modes and horizons of tasks?
2. Was a human baseline tested to contextualize the ~40% model accuracy?
3. Could the framework be extended to sequential reasoning or time-evolving dashboards (e.g., live updates, streaming data)?

**Details Of Ethics Concerns:**

No.

---

### Official Review · Reviewer_wTgB · 2025-10-27

**Soundness:** 2
**Presentation:** 2
**Contribution:** 2
**Rating:** 4
**Confidence:** 4

**Summary:**

The paper proposes VizAgentBench, a benchmark for interactive dashboard understanding where agents must act on coordinated multi‑view visualizations to answer questions. It comprises 25 templates over five chart types, yielding 192 tasks from 10 datasets, with a dual mouse/command interface that separates perception from action. Across 4 frameworks and 7 MLLMs, the best Pass@1 is ~44%, revealing substantial difficulty and recurrent errors in perception, memory, and planning.

**Strengths:**

1. The paper evaluates various agent frameworks and analyzes the impact of the framework design on performance.
2. Multiple chart types and MCV metrics are incorporated to ensure a more comprehensive and realistic evaluation.

**Weaknesses:**

1. The contributions of the paper seem limited - the paper does not propose clear new insights and just applies a variety of MLLMs on the MCV tasks. The benchmark doesn't seem to offer any insights for improvement (better agent framework or model improvement).

2. There is a lack of human baselines. It would be helpful to see how well humans perform on these problems for comparison.

3. There may be bias in question generation. Since the questions are produced by GPT-4.1, they are likely to be more aligned with the knowledge and preferences of OpenAI models.

4. The benchmark has not been released, and the paper provides no example questions or prompts for reference. I am also concerned about the quality of the tasks, as the example in Figure 3 is poorly presented - part of the legend overlaps with the pie chart, and the text contains incorrect capitalization.

**Questions:**

1. Can you provide a more in-depth analysis or case study on the impact of different agent frameworks? For example, why does OpenManus outperform ReAct and AutoGen?

2. Are 10 interaction turns sufficient to solve the problems? What happens if more turns are allowed?

3. In the final results, are both “Mouse Interaction” and “Command” actions available to agents? If so, could you show the agents’ action preferences? Would the results improve if the agent were restricted to using only one type of action?

4. It's helpful to see how reasoning/thinking improves the agent performance. Can you do experiments/analysis around it?

5. Can you provide the breakdown of results for 2-view, 3-view, and 4-view MCVs? Do agents struggle more as the number of views increases?

6. What is the token consumption for running the evaluation?

7. Figure 3 appears quite unprofessional, as the text in the screenshot is cut off at the edge.

---

### Official Review · Reviewer_LUjd · 2025-11-01

**Soundness:** 3
**Presentation:** 3
**Contribution:** 3
**Rating:** 2
**Confidence:** 4

**Summary:**

The paper contributes by introducing VizAgentBench, a benchmark for evaluating multimodal agents on interactive, multi-view visual analytics (MCVs), which addresses the limitation of existing static, single-view benchmarks, and by constructing 192 dashboards (with QA tasks) and evaluating 7 SOTA MLLMs across 4 frameworks.

**Strengths:**

1. Originality: VizAgentBench targets the intersection of multimodal agents and interactive MCVs, an area ignored by prior GUI benchmarks and chart QA datasets. The coordination logic ensures the benchmark reflects real-world analytical workflows.
2. Quality: The benchmark is diverse (10 datasets across finance/healthcare/sports, 25 templates, 192 tasks) and rigorous (manual QA validation, controlled experiments across models/frameworks).
3. Clarity: The paper effectively communicates complex concepts (e.g., MCV coordination, agent-environment interaction) through simple examples and visual aids. The experimental results are transparent, with Table 6 providing granular data on model-framework pairs to enable comparison.
4. Significance: The low accuracy of top models (~44% Pass@1) demonstrates a critical gap in current agents, while the case study identifies specific failures (visual recognition, context memory) that prioritize future research. This work will likely become a standard benchmark for evaluating agent performance on visual analytics.

**Weaknesses:**

1. After submission, the paper does not provide any code or data, whether as supplementary materials or via an anonymous link. This impedes the early validation of its key components, a critical aspect of the essential requirements for a benchmark-focused paper.
2. While the 25 templates are "manually authored", they lack validation against real-world dashboards. For example, the paper fails to confirm whether the distribution of templates (such as 4-view MCVs, as shown in Table 4) aligns with the frequency of view counts in industry tools like Tableau or Power BI. This casts doubt on whether the agent performance measured by VizAgentBench can be generalized to real-world dashboards.
3. The case study, e.g. in Section 5.3, outlines specific failures but does not analyze the distribution of these failure patterns, avoiding deeper exploration and research.
4. The paper does not report human accuracy on VizAgentBench tasks. Without this baseline, it is impossible to judge whether the ~44% accuracy of top models stems from a fundamental limitation of current AI or an excessively complex benchmark, one that even humans would struggle with.

**Questions:**

Address the issues in the Weaknesses Section.

---

### Official Review · Reviewer_WDAr · 2025-11-01

**Soundness:** 3
**Presentation:** 3
**Contribution:** 3
**Rating:** 4
**Confidence:** 4

**Summary:**

This paper presents VizAgentBench, a benchmark for evaluating multimodal language model (MLLM) agents on interactive, multi-view visualization reasoning. Unlike previous static chart QA datasets, VizAgentBench requires agents to manipulate coordinated dashboards through actions like filtering and brushing to answer analytical questions. Built from 10 real-world datasets and 25 visualization templates, it includes 192 dashboard–question pairs. Experiments on seven MLLMs and four agent frameworks reveal that even leading systems (e.g., GPT-5) achieve only around 44% on Pass@1, highlighting major challenges in perception, memory, and reasoning for visual analytics agents.

**Strengths:**

- Novelty: Introduces the first benchmark explicitly targeting interactive, multi-view visual reasoning by agents.

- Technical soundness: Provides a clear generation pipeline, coordination logic, and reproducible environment.

- Comprehensive evaluation: Compares multiple agent frameworks and leading MLLMs.

- Insightful findings: Identifies key failure modes such as lack of visual memory and unstable multi-turn reasoning.

- Open-source impact: The proposed API design (separating perception from action) can accelerate follow-up research.

**Weaknesses:**

- Scale and coverage: Only 192 dashboards and questions may not ensure sufficient diversity or statistical robustness.

- LLM generation bias: Since GPT-4.1 generated all questions, potential bias favoring OpenAI models should be addressed through cross-model verification or ablation.

- Limited quantitative validation: Manual inspection is described but lacks measurable inter-rater reliability or coverage statistics.

- Evaluation metrics: Accuracy and Pass@k alone may not fully capture interaction efficiency or reasoning depth. Fine-grained metrics such as per-category pass rates, reasoning chain length, or time cost would improve analysis.

- Interaction settings: The “10-turn limit” seems arbitrary without evidence that it fits typical reasoning complexity.

- Incomplete error analysis: The qualitative case study could be expanded with statistical analysis of all failed cases.

**Questions:**

- How do the authors ensure that GPT-4.1’s question generation does not systematically advantage OpenAI models during evaluation?

- Can agents combine the two interaction modalities (mouse and command) within a single task?

- How was the 10-turn interaction limit chosen, and how sensitive are results to this parameter?

- Could additional efficiency metrics (e.g., average turns to success) be reported?

- Would a “visual memory” module improve performance in multi-hop reasoning?

---

### Meta-Review · Area_Chair_FrZL · 2026-01-09

**Summary:**

Reviewers recognize that the paper proposes a new benchmark for evaluating multimodal agents on interactive, multi-view visualization tasks and highlights the difficulty of such settings for current systems **[WDAr, LUjd, wTgB, aEAe]**.
The benchmark construction and evaluation across multiple agents and models are acknowledged as a reasonable first step beyond static chart QA.

However, reviewers raise substantial concerns about benchmark scale, realism,and evaluation design.
These include the small number of tasks, lack of human baselines, potential bias from GPT-4.1-generated questions, unclear interaction constraints, limited metrics, and insufficient analysis of agent behavior and failure modes **[WDAr, LUjd, wTgB, aEAe]**.
Several reviewers also question whether the benchmark provides clear methodological insight beyond evaluating existing agents **[LUjd, wTgB]**.
The authors did not participate in the rebuttal, leaving these concerns unaddressed. Reviewer consensus remains below the acceptance threshold.

**Reviewer Concerns:**

Addressed by the rebuttal
- None. The authors did not participate in the rebuttal.

Outstanding
- Benchmark scale and diversity ([WDAr, LUjd, wTgB])
- Lack of human baselines ([LUjd, wTgB, aEAe])
- Question generation bias ([WDAr, wTgB])
- Evaluation metrics and interaction constraints ([WDAr, wTgB, aEAe])
- Limited analysis and insight ([LUjd, wTgB, aEAe])
- Reproducibility and release status at review time ([LUjd, wTgB])

**Reviewer Scores:**

All reviewers provided below-threshold or marginal scores, and there was no rebuttal or discussion to suggest any change.

---

### Decision · Program_Chairs · 2026-01-26

Reject